# A Clinical Efficacy of PRRT in Patients with Advanced, Nonresectable, Paraganglioma-Pheochromocytoma, Related to SDHx Gene Mutation

**DOI:** 10.3390/jcm8070952

**Published:** 2019-06-30

**Authors:** Agnieszka Kolasinska-Ćwikła, Mariola Pęczkowska, Jarosław B. Ćwikła, Ilona Michałowska, Jakub M. Pałucki, Lisa Bodei, Anna Lewczuk-Myślicka, Andrzej Januszewicz

**Affiliations:** 1Department of Oncology and Radiotherapy and Department of Radiology, Maria Skłodowska-Curie Memorial Cancer Center, 02-034 Warsaw, Poland; 2Department of Hypertension, and Department of Radiology Institute of Cardiology, 04-628 Warsaw, Poland; 3Department of Cardiology and Cardiosurgery, School of Medicine, University of Warmia and Mazury, 10-082 Olsztyn, Poland; 4Molecular Imaging and Therapy Service, Department of Radiology, Memorial Sloan Kettering Cancer Center, New York, NY 10065, USA; 5Department of Internal Medicine and Endocrinology, Medical University of Gdansk, 80-211 Gdansk, Poland

**Keywords:** paraganglioma/pheochromocytoma (PPGL), peptide receptor radionuclide therapy (PRRT), SDHx genes mutation

## Abstract

Paragangliomas and pheochromytomas (PPGLs) exhibit variable malignancy, advanced/hormonally active/progressive need therapy. PRRT (Peptide Receptor Radionuclide Therapy) could be an option for these patients. To evaluate the effectiveness of PRRT (90Y DOTATATE), based on overall survival (OS) and progression-free survival (PFS), in patients with PPGLs, related to SDHx gene mutation, we conducted a prospective open-label, single-center, phase II study. Thirteen patients were observed, eight PGL1 and five PGL4, all with advanced, non-resectable tumors, and eight had metastases. All were treated with 90Y DOTATATE. Efficacy was based on OS and PFS, and radiological response was based on RECIST. Hormonal activity was evaluated using serum-fractionated free catecholamines. Eight subjects had a clinical response, three were stable, and two exhibited disease progression. Among four patients with hormonally-active PPGLs, three showed a reduction and one showed normalization. OS for all was 68.0 months; PFS was 35.0 months. OS in PGL4 = 25.0 vs. N.R. (not reached) in PGL1. PFS in PGL4 = 12.0 vs. N.R. in PGL1. A difference was seen in the OS and PFS in patients who did not respond clinically, compared to those who did, OS = 22.0 vs. N.R. PFS = 7.0 vs. N.R. A difference in the OS and PFS was noted in patients with liver and bone involvement compared to those without. PRRT is an effective therapy in selected population of patients with SDHx, in those with locally-advanced, non-resectable tumors. Furthermore, it is effective regardless of the secretory status.

## 1. Introduction

Paraganglioma (PGL) and pheochromocytoma (PCC), commonly called PPGLs, are rare, biologically and clinically heterogeneous neuroendocrine tumors. Paragangliomas arise from the sympathetic/parasympathetic chain ganglia. Pheochromocytomas arise from adrenal medulla chromaffin cells. The incidence of PPGL based on nearly 1500 patients who were diagnosed and histopathologically confirmed between 1995 and 2015 was 0.04–0.21 per 100,000 person-years [1]. PPGLs can be found within the parasympathetic and sympathetic autonomic nervous system from the skull base to pelvis. Sympathetic tumors often secrete catecholamines [2,3] and are usually associated with hypertension and other symptoms, typically including palpitations, sweating, pallor, headache, and anxiety, but also life–threatening cardiac complications, such as hypertension crisis, different types of cardiomyopathies, life-threatening arrhythmias, end-organ damage, cardiogenic shock, etc. [3]. 

Despite of cardiovascular complication PPGLs exhibit variable risk of malignancy, about 15–20% of these tumors are metastatic [4]. There are no histological, genetic, or molecular markers that could distinguish between benign and malignant disease and, subsequently, the diagnosis of malignancy relies exclusively on the presence of metastases. Since all PPGLs have some metastatic potential, the terms metastatic and non-metastatic pheochromocytoma are, therefore, preferred over the use of benign and malignant [5,6]. Metastases usually involve the lymph nodes (80%), the skeletal tissue (71%), the liver (50%), and the lungs (50%) [7,8]. 

PPGLs are also recognized to have the highest degree of heritability amongst any neuroendocrine tumors. There are more than 20 different genes with both germline or sporadic driver mutations and an increasing number of potential diseases modifying genes. Transcriptomic profiling has identified four clusters of pathogenic pathways involved in tumorigenesis in PPGL [9,10].

Cluster 1 tumors develop in patients with germline or somatic mutations in *VHL*, *SDHB, SDHD*, *SDHC*, *SDHAF2*, *SDHAF2*, *EPAS1*, *PHD2*, *MDH2*, and *FH* genes and involve activation of hypoxia-angiogenic pathways. This cluster is subsequently subdivided into: the tricarboxylic acid (TCA) cycle-related oncometabolite accumulation group, consisting of *SDHx* genes and the *VHL*/*EPAS*-related group, associated with the direct disturbance in HIF (hypoxia inducible factor) turnover [9,10]. Cluster 2 tumors develop in patients with mutations in *RET*, *NF1*, *TMEM127*, and *MAX* and involve RAS and kinase signaling pathways. Cluster 3 includes Wnt signaling PPGL. It is associated with somatic mutations in *CSDE1* or somatic gene fusions *UBTF-MAML3* that cause activation of the Wnt receptor signaling and Hedgehog signaling pathways [9]. Cluster 3 tumors develop in patients with mutations in *RET, NF1, TMEM127*, and *MAX* and involve RAS and kinase signaling pathways. Finally, cluster 4, the cortical admixture subtype overexpresses known adrenal cortex markers (CYP11B1, CYP21A2, and StAR) and PPGLs markers and is associated with mutations in *MAX* [9].

Germline mutations in succinate dehydrogenase enzyme complex (*SDHx*) genes are one of the most common genetic causes of PPGLs, occurring in up to 25% cases [8,11,12,13]. 

There are five paraganglioma-pheochromocytoma syndromes associated with heterozygous germline mutations in genes encoding the subunits of the succinate dehydrogenase enzyme complex: PGL1–*SDHD* gene; PGL2–*SDHAF2* gene; PGL3–*SDHC* gene; PGL4–*SDHB* gene; and PGL5–*SDHA* gene. Among them the most common are PGL1 and PGL4 [12]. 

Mutations in succinate dehydrogenase (*SDHx*) predispose patients to aggressive phenotypes, such as distal metastasis, in some series in up to 70% of patients, and tumor multiplicity and recurrence, which emphasizes an urgent need for effective therapies against *SDHx*-mutated PPGLs [8,12,13,14,15,16,17,18,19].

Each transcriptomic cluster mentioned above has a unique molecular-clinical-biochemical-imaging phenotype which can be used in personalized management involving diagnostics, targeted therapy, and follow-up. First, these pathogenic differences are associated with different hormonal phenotypes: cluster 1 tumors excrete mainly noradrenaline while cluster 2 and 3 tumors are mixed adrenergic/noradrenergic where adrenaline excretion dominates. Furthermore, cluster 1 pseudohypoxic TCA cycle-related PPGLs show a relatively low cell differentiation, often with a decreased or absence of key enzyme in catecholamine metabolism what results in purely noradrenergic and/or dopaminergic phenotypes together with a lower overall content of catecholamines. It means that pseudohypoxic TCA cycle-related PPGLs cause less catecholamine-related symptoms than other subgroups of PPGL, but it also means that the disease may be unrecognized for a long time and that these patients may reach more advanced stages at diagnosis [9,20,21,22,23].

Secondly, the somatostatin receptor (SSTR) 2A is also expressed heterogeneously in PPGL. It has been recently showed that especially those PPGLs of the TCA cycle-related pseudohypoxic subtype, have significantly high expression of SSTRs. This phenomenon can be used in theranostic application of radionuclides targeted SSTR especially in patients with multifocal, non-resectable, and/or metastatic, progressive disease [9,24,25,26].

In the past, most popular oncologic treatment in this group of patients was: ^131^I-meta-iodobenzylguanidine (mIBG), particular for tumors demonstrating high uptake of ^123^I mIBG and chemotherapy based on Averbuch protocol with cyclophosphamide, vincristine, and dacarbazine (CVD), but these approaches usually fail to produce a prolonged remission and are associated with relatively significant marrow toxicity [27,28,29,30,31,32,33].

Alkylating agent temozolomide (TMZ), which has a similar safety profile and effects with dacarbazine showed a 50% partial response rate in SDHB carriers (*n* = 10) in a retrospective series which suggest that TMZ might be more effective against metastatic PPGLs in PGL4 syndrome [34].

New therapeutic options based on molecular pathways leading to tumorigenesis in PPGLs which include biological agents, such as tyrosine kinase inhibitors (TKIs) have revealed anti-neoplastic effects but also limited effectiveness [35,36,37]. 

Based on the guidance from the distinctive transcriptomic profile, new therapeutic agents are now under investigation in PPGLs. Future perspectives concern the use of drugs interfering with different pathological pathways leading to neoplastic transformation in PPGLs, such as those associated with distinct features of methylome, metabolome, and hypoxiome [9].

In addition to all these therapeutic options, there are another molecular targeted therapies, which have or may have a positive impact in metastatic or unresectable PPGLs. These therapies include radiopharmaceutical medications, such as ^131^I-mIBG and peptide receptor radionuclide therapy (PRRT). There are generally limited data on the result of PRRT treatment in PPGLs in the literature. Preliminary experience suggests low toxicity and favorable efficacy in disease control were reported by several researchers [38,39,40,41,42]. 

Only two reports concerned outcome of PRRT in patients with exclusively SDHx mutations; Zovato et al. described four patients with PGL1 and head and neck paragangliomas [38], Pinato et al. described five patients with PGL4 and metastatic PPGLs [40]. Our series consists of 13 subjects with PGL1 and PGL4, all with metastatic or unresectable PPGLs is so far the largest one. 

It is extremely important to recognize that different cluster biomarkers may be responsible for the clinical response or resistance to therapy and clinicians involving in the treatment of metastatic or inoperable PPGLs should try to match them with choice of potentially the most superior intervention [9].

Since (TCA) cycle-related PPGLs show significantly high expression of SSTR, the aim of this study was to evaluate the clinical effectiveness of PRRT using (^90^Y DOTA0, D-Phe1, Tyr3-octreotate), based on overall survival (OS) and progression-free survival (PFS) in 13 patients with PGL1 and PGL4 syndromes and unresectable or metastatic PPGLs treated in a single center. 

## 2. Materials and Methods 

This was a prospective, interventional, single institution open-label, phase II study, which was approved by the Clinical Ethics Committee of the Institution where the study was conducted. 

Research of Ethics Authors declare that the investigations were carried out following the rules of the Declaration of Helsinki of 1975 (https://www.wma.net/what-we-do/medical-ethics/declaration-of-helsinki/), revised in 2013, All subjects included in this study have given their written informed consent for inclusion before they participated in the study. The study protocol has been approved by Clinical Ethics Committee of the Institution where the study was conducted—Central Clinical Hospital of Ministry of Interior Affairs and Administration. Komisja Etyki I Nadzoru nad Badaniami na Ludziach Centralnego Szpitala Klinicznego MSWiA 02-507 Warszawa; Wołoska 137; Poland. Date and number of approval: 03.11.2004 No 109/2004.

The time frame of the study: it was conducted between December 2006 and January 2018. Prior to study inclusion all patients understood the experimental nature of this type of treatment and provided their written consent. During time frame of the study, which included, overall, 156 patients with various types of neuroendocrine neoplasms (NEN), we selected a group with germline mutation SDHx and histological diagnosis of PPGLs in the current study.

### 2.1. Patients 

The group comprised of 13 patients, including five females (42%), with a mean age of 41.8 years (range: 27–62 years). The inclusion criteria were histological diagnosis of PPGLs and confirmed germline mutation in SDHD or SDHB genes; adults ≥ 18 years old, male or female, performance status (PS) 0–2; adequate renal, bone marrow, and liver function; and life expectancy of at least three months. The tumor parameters can be measured objectively as the size to be assessed in radiological studies CT/MRI used the RECIST 1.0 criteria. 

All patients had high over-expression of SST receptors, seen in somatostatin receptor scintigraphy (SRS), performed before PRRT, using ^99m^Tc-[HYNIC, Tyr3]octreotide (TOC) (Tektrotyd^®^, National Centre for Nuclear Research—Polatom, Poland) with radiotracer avid lesions seen in any part of the body and having uptake at least or greater than physiological liver activity (Krenning score of at least 2) [43]. 

The major exclusion criteria were as follows: (1) uncontrolled metastases to the central nervous system, (CNS); (2) poorly controlled concurrent medical illness; (3) symptomatic heart failure NYHA class III or IV, congestive cardiac failure, myocardial infarction in the last six months, serious uncontrolled cardiac arrhythmia, unstable angina; (4) active uncontrolled infection (5) Pregnant, or breast feeding female patients; (6) exclusive clinical and laboratory findings that may compromise the patient’s safety; and (7) the presence of any psychological, familial, sociological, or geographical condition potentially hampering compliance with the study protocol and follow-up schedule, including alcohol dependence or drug abuse. 

All subjects had advance, non-resectable, progressive disease, four of them with hormonally-active hypersecretion. Clinical, biochemical and/or imaging progression (CT, MRI or SRS) were seen in all of them. This was assessed for a minimum period of three months prior to treatment. Patients’ symptoms and adverse events evaluated using Common Terminology Criteria for Adverse Events (CTCAE NCI), Version 4.0 [44].

It was performed after each cycle and then six weeks after completed PRRT followed by 3 months visit after the completion of treatment courses. Routine hematology, liver, renal, as well as tumor markers chromogranin A (CgA) and serum fractionated free catecholamines) were performed before each therapy, as well as at follow-up visits. 

All subjects had structural imaging (CT or MRI) performed between four and six months after the completion of PRRT, and then repeated at six month intervals during clinical follow-up. Tumor response to treatment was evaluated according to RECIST 1.0. All images were reviewed by an experienced radiologist (IM).

### 2.2. Study Parameters

PRRT was performed by i.v. ^90^Y DOTATATE administration in all subjects. Active therapy using “cold” somatostatin analogues, if present, was discontinued at least four weeks before the start of PRRT. Exclusion criteria were as follows: Hb < 8 g/dl, WBC <2 × 10^3^/μL, platelets < 100 × 10^3^/μL, creatinine level > 2 mg/dL or GFR < 30 mL/min and poor performance status (Karnofsky score below 50 or EOCG status 3 or 4), which were the same as mentioned previously in the GEP-NET group [45]. Contraindications included pregnancy, or established myelosuppression, or significant renal failure, mentioned above.

### 2.3. Preparation of Radiotracer

Labelling: 100 µg DOTA-D-Phe1-Tyr3-octreotate was labelled with max. 7.5 GBq ^90^Y in not more than 0.5 mL in 0.05 M HCl (POLATOM, Poland) incubating at 95 °C for 25 min. In all chromatography runs radiochemical purity was >99.0%, with no need for purification. The whole proceedings of labelling were mentioned previously [45]. Immediately before therapy the ^90^Y-DOTATATE with a mean activity per session of 3.7 GBq was diluted in saline to a final volume of 50 mL and given via i.v. to patients using an infusion pump 

### 2.4. Therapy–Administration Protocol

Treatment administration protocol: Ondansetron (8 mg; GlaxoSmithKline, London, UK) was administered by i.v. 30 min before amino acid (AA) administration. The AA infusion was to prevent the development of delayed renal toxicity. The slow intravenous infusion of AA (1500 mL; Vamin 18 and Nephrotect; Fresenius Kabi, Bad Homburg, Germany) were carried out for 1.0–1.5 h, then ^90^Y DOTATATE was added via an infusion pump system with a mean speed of 150 mL/h. and was continued for a approx. 20–25 min. Every subject was treated at intervals of 8–12 weeks. The treatment interval could be extended to 16 weeks in patients with longer continuing subacute hematologic adverse events (AEs). All subjects with at least two sessions of PRRT were included in the study. 

### 2.5. Biodistribution of the Radiotracer and Dosimetry

In order to confirm that the therapeutic uptake of the radiotracer had a similar biodistribution to the diagnostic somatostatin receptor scintigraphy (SRS), post-therapy “Bremsstrahlung” imaging was used in each case. All scans were acquired from 6–12 h after administration of ^90^Y DOTATATE using a dual-head gamma camera equipped with medium-energy all purposes collimation (MEAP) with a photopeak centered on 95 keV with a 50% window (e-cam, Siemens Healthcare, Hoffman Estates IL 60195; USA). In postprocessing we routinely utilized 64 projections (a 64 × 64 matrix), 20 s per projection with no zoom. Reconstruction algorithms were based on commercially available iterative reconstruction software, i.e., OSEM, including four subsets and four iterations with a standard Gaussian filter - 3D-flash (Siemens Healthcare, Hoffman Estates IL 60195; USA).

Due to the pure beta emission of ^90^Y, internal dosimetry was evaluated based on previous studies using ^90^Y-DOTATOC treatment [46,47] and carried out using the Medical Internal Radiation Dose method. Using the cumulative activity as described above, the expected radiation dose to bone marrow was <2 Gy, and to the kidneys <23 Gy. 

### 2.6. Radiology Proceedings

Each case, except for a single subject, had a multidetector CT, including the head and neck area, then chest, abdomen, and pelvis. In each case, CT was performed after i.v. contrast enhancement, approx. 100–120 mL (1.4 mg/kg) of low ionic contrast medium intravenously, at a rate of 3.5–5.2 mL/s. The time of the acquisition after i.v. contrast enhancement was as follows: 25 s, 50 s, and 240 s, respectively. The spiral CT acquisition through the body was accomplished during a breath-hold moment. A standard 512 × 512 matrix was used in each case. The slice thickness was 1 mm, which was then used to produce reformatting images using the MIP, MPR techniques, and 3D reconstruction. A transverse coronal and sagittal projection were used in every case, considering image analysis. CT images were interpreted using a dedicated CT workstation, with total freedom for window and level adjustments, and for the magnification of each image at the time of the analysis. 

### 2.7. Assessment of Efficacy 

Clinical response and performance status: Clinical response to PRRT based on physical performance status (PS) and quality-of-life questionnaires (QLQ) was assessed before treatment, six weeks after completion of the therapy and then at three-month intervals. The patients were assessed for clinical responses by completing a self-assessment questionnaire (European Organization for Research and Treatment of Cancer EORTC, QLQ–C30) during physician directed interviews carried out before the beginning of treatment, next before each treatment cycle and then followed by three monthly intervals. The items they assessed included appetite, malaise, weight loss, and the presence, intensity, and frequency of any pain related to disease, nausea, vomiting, fever, wheezing, and abdominal bloating. Any analgesic and others drug requirements before and after treatment were recorded. 

The patients’ clinical PS was assessed before six weeks and six months after the completion of therapy using standard ECOG classification by four observers (AKC, MP, ALM, JBC). Routine complete blood count and liver and renal function tests were performed before and three weeks after each therapy session and at follow-up visits at six weeks, three months, and six months after the last treatment cycle and thereafter at six-month intervals. The evaluation of objective response in each case was utilized by multiphase structural imaging (CT/MRI), described above. The radiological response was based on RECIST 1.0 using standard terminology of objective response, performed at six months after the PRRT and then followed be intervals of six months during first two years of follow-up, and after that, annually. 

Biochemical tumor response. The biochemical tumor response was determined by serial measurements of plasma CgA. The CgA was measured using Chromoa^®^ Chromogranin A kit (ALPCO, Salem NH, USA). The upper limit of normal was 102.0 ng/mL. Plasma-free normetanephrine, metanephrine and methoxytyramine measurements were performed using the UltiMate 3000 HPLC system equipped with a Coulochem III detector (both from Thermo Fisher Scientific, Waltham, MA, USA), according to a previously-described HPLC-ECD method, for which lower limits of quantification had been established at 10 pg/mL for all the analytes of interest [48].

### 2.8. Outcomes

The primary endpoint was overall survival (OS), defined as the time between enrolment into the study and any related to disease death. Secondary endpoints included progression-free survival (PFS), defined as the time from the date of enrolment to the date of disease progression (DP), or death, along with clinical and radiological response rates. Clinical response rates were based on performance status (ECOG/Karnofsky), which was assessed independently by four observers prior to therapy, at six weeks, and then six months after completing the treatment and at subsequent half-yearly intervals (AKĆ, MP, ALM, JBĆ). Any adverse events were recorded according to the Common Terminology Criteria for Adverse Events (CTCAE), Version 4.0 [44].

### 2.9. Statistical Analysis

The normality of continuous variable distribution was verified with the Kolmogorov–Smirnov test. The statistical characteristics of continuous variables were presented as medians and interquartile ranges; the Wilcoxon matched pair test was used to compare these variables between the groups of patients before and after therapy. The distributions of discrete variables were compared with Pearson’s chi-square test or Fisher’s exact tests. 

OS and PFS were evaluated using the Kaplan-Meier method, the differences between groups in PFS and OS were calculated using Cox–Mantel test. All statistical calculations were carried out using the Statistica 12 software package (StatSoft, Tulsa, OK, USA), with the level of statistical significance set at *p* < 0.05.

## 3. Results

Summarized demographic, clinical, pathological, and mutation data, including localization of primary tumors all patients presented in Table 1, Table 2 and Table 3. Overall 32 therapeutic sessions were performed in a group of 13 patients with *SDHx* mutations, with a mean cumulative administered activity of 8.3 GBq of 90Y per patient (range 6.4–14.8 GBq), mean 3.4 GBq per therapy session. A total of eight patients received only two cycles of treatment. Details of the given activity in the group of *SDHx* patients, and also separate in *SDHB* and *SDHD* subjects, are presented in Table 4.

Initially, four subjects presented with secretor tumors, with elevated free plasma catecholamines, three of them exhibited a reduction of free plasma catecholamines and one presented a normalization three months after PRRT (Table 2). During and after PRRT subjects had significant clinical symptoms and signs due to catecholamine oversecretion. 

Overall, 32 therapeutic sessions were performed in a group of 13 patients with SDHx mutations, with a mean cumulative administered activity of 8.3 GBq of 90Y per patient (range 6.4–14.8 GBq), mean 3.4 GBq per therapy session. A total of eight patients received only two cycles of treatment. Details of given activity in the group of SDHx patients and also separate in the SDHB and SDHD subjects are presented in Table 4.

### 3.1. Clinical Response

After six weeks, and then at six months after PRRT, eight patients had clinical response, three had stable disease (SD), and two had disease progression (DP), based on the before and after PRRT clinical stage (CS) evaluated by performance status (PS) (ECOG/Karnofsky) (*p* < 0.05 Wilcoxon matched pair test). In all patients with secretor tumors (two subjects with SDHD and two with SDHB), we observed early hormonal response. In only one patient with moderate plasma normetanephrine elevation before, and normalization after PRTT we were able to withdraw all antihypertensive drugs. The remaining three patients initially had a significantly elevated level of plasma metanephrines, which only slight decreased after PRRT, and the antihypertensive treatment after PRRT consisted of the same medications in the same doses. There was no significant difference in OS and PFS between those with secretor tumors compared to non-secretor tumors (Table 5).

### 3.2. Radiological Response (ORR)

Radiological response was assessed in 12 patients, eight with PGL1, and four with PGL4. One SDHB mutation carrier with a urinary bladder tumor and bone metastases had only a single PRRT treatment session and then developed a pathological spine fracture, which led to tetraplegia. After the fracture, the patient was unable to reach the next dose of PRRT and died 12 months later. Since the study was initiated in 2006, when RECIST 1.0 were the utilized criteria, this approach was maintained in each case during clinical follow-up. After six months, partial response (PR) was reported in one subject (8%), stable disease (SD) in nine (75%), and disease progression (DP) in two (17%). After 12 months of follow-up, there was no PR, nine subjects (82%) had SD and two patients (18%) had DP. Detailed ORR in the whole group and in the subgroups with PGL1 (SDHD) or PGL4 (SDHB) are presented in Table 6.

### 3.3. Outcome

Median OS for all subjects was 68.0 months (CI −/+95% 38.6–105.1), PFS 35.0 months (CI −/+95% 24.4–93.1) (Figure 1). During long-term clinical follow-up, eight patients (62%) exhibited disease progression and six (46%) perished. OS and PFS were not significantly different among males and females, 63.5 vs. 68.0 months and 39.5 vs. 35.0 months, respectively (Cox Mantel test *p* > 0.05). 

Significant differences in median OS were noted between patients with PGL1 compared to subjects with PGL4: N.R. vs. 25.0 months (*p* = 0.05) (Figure 2). The same was noted in PFS, which was N.R. vs. 12 months (*p* = 0.014) (Figure 3). All patients with PGL4 progressed, as opposed to only three in PGL1 group. In the PGL4 group there were four (80%) deaths, compared to only two (25%) in PGL1. 

There was a significant difference in OS and PFS in patients who responded clinically, based on improvement of PS, to PRRT (evaluated after six weeks), compared to those who exhibited stability or progression: median OS was N.R. vs. 22.0 months (*p* = 0.005) (Figure 4) and median PFS was N.R. vs. 7.0 months (*p* < 0.001) (Figure 5). 

There was also a significant difference in patients without liver involvement, compared to those with metastatic liver disease: median OS was N.R. vs. 25.0 months (*p* = 0.033) and median PFS was N.R. vs. 10.0 months (*p* = 0.003). A similar difference was noted when comparing patients without and with bone metastases: OS was N.R. vs. 25.0 months (*p* = 0.03) and PFS was N.R. vs. 12.0 months (*p* = 0.003). Summarized details of median OS and PFS in different groups of subjects are presented in Table 5. 

Example of patient with metastatic disease is presented in (Figure 6).

### 3.4. Adverse Events CTC AEs NCI v. 4.0

No major acute adverse events were recorded either during the therapy sessions or within 24 h of the ^90^Y-DOTATATE infusion. Mild nausea was noted in three patients (23%), and vomiting in one, probably related to the co-administration of amino acid infusion. These effects disappeared spontaneously within 2 h of discontinuation of the amino acid infusion.

In four patients with hormonally active tumors there were no adverse events during and immediately after the infusion of ^90^Y DOTATATE. No other signs and symptoms of hypersecretion of and significant adverse reaction were noted. In single cases, there was abdominal cramping pain requiring analgesics, while one patient complained of headache. All of the abovementioned were recorded in only a single cycle and the symptoms did not represent in the subsequent cycles. All other patients required less than 12 h hospital admission, and all patients were treated as outpatients.

### 3.5. Renal Toxicity

During the therapy, and up to six months thereafter, no significant renal toxicity was observed. Twelve months after the completion of therapy, two patients exhibited grade 2 renal toxicity (according to CTCAE 4.03). After 18 months of follow-up, one patient developed grade 3 toxicity and another one exhibited grade 3 toxicity after 36 months. The first one had previously received extensive nephrotoxic chemotherapy and radiotherapy of the pelvis due to massive bone metastasis, finally required dialysis, also both had rapid disease progression.

### 3.6. Hematological Toxicity

Grade 3 anemia was noted in the same two patients who developed renal toxicity. In one case after 12 months after the end of therapy in subject previously treated with chemotherapy and radiotherapy, in second after 24 months without previous chemotherapy. There was no other hematological toxicity in the rest of the patients. In our study, we noted a transient grade 1–2 lymphopenia, with a nadir occurring 20 days after each cycle of ^90^Y DOTATATE infusion.

## 4. Discussion

The complex nature of PPGLs where the site of origin and genotype may influence their clinical behavior and prognosis makes these tumors more difficult to treat than other cancers. The endocrine nature of the disease, the elevated risk for severe cardiovascular disease, and the risk of malignancy represent further challenges. Surgery is the gold standard, but even after radical operation patients need long-term follow-up due to the risk of recurrence or development of metastases, even after many years. The real challenge for clinicians is the treatment of inoperable or metastatic tumors [1,2,3,4].

Herein, we present the outcomes after PRRT in 13 patients with *SDHD* and *SDHB* gene mutations. We believe that this is, for some reason, a unique group of patients. First and foremost, this is a homogenous group of PPGLs with known genetic background. The molecular genetics and in consequence “omic” approach provided a powerful tool which has utility in the identification of new therapeutic and potentially diagnostic targets. It was previously mentioned that PPGLs are heterogeneous with respect to the expression of SSTR [23,24,25,26].

Once more, pseudohypoxic TCA cycle-related PPGLs were shown to be unique, having a relatively high expression of SSTR type 2 which allows supposing that molecular treatment targeted SSTR will be particularly effective in this distinctive group [23,24].

Very few studies have addressed the effect of PRRT in the management of patients with metastatic or unresectable PGLs and PCCs. However, all showed low toxicity and efficacy in disease control. In 2006, Van Essen et al. published a report concerning the use of ^177^Lu-DOTATATE in 12 patients with metastatic PGLs. Two of four patients with progressive PGL had tumor regression and one had stable disease (SD). Among the five patients with stable PGL, two had SD, two had progressive disease (PD), and in one patient treatment outcome could not be determined. PGL was stable in three patients whose disease status at the beginning of therapy was unknown [42]. These results are similar in terms of ORR with our results, presented above.

Forrer et al [39] reported an overall response to treatment in 22 of 28 (78%) patients, with a PFS ranging from three to 43 months, which seems to be similar to our results, consider median PFS (35.0 months, CI 24.4–93.1). Prasad et al. [49] reported the results of PRRT (^90^Y-DOTATATE, ^177^Lu-DOTATATE, and combination) in 20 patients with progressive PGLs and PCCs. They reported an 80% and 43% overall response rate with ^177^Lu-DOTATATE and ^90^Y-DOTATATE treatment, respectively.

The result favored the use of ^177^Lu DOTATATE over ^90^Y DOTATE. Another single case study published by Cecchin et al. in 2011 described the use of ^177^Lu-DOTATATE in a case with multiple spinal canal and cranial PGLs with a mean volume reduction of 70%, which could be used in this clinically difficult case with intradural PGLs [50].

The first report in hereditary PGL1 patients treated with ^177^Lu DOTATATE was published by Zovato, wherein four subjects with HNP and mediastinal PGLs were treated successfully, all with hereditary PGL1. All of them had a partial response or a stable disease after the treatment [38]. Puranik et al. reported the outcome of PRRT in a series of nine patients with inoperable head and neck paragangliomas [41]. They demonstrated disease stabilization and symptomatic relief of these patients after treatment, which is in agreement with our results in those patients with HNPs who remained stable during the long-term follow up [41]. 

Pinato et al. [40] reported safety and efficacy outcomes from a case series of five patients with advanced PPGLs treated with ^177^Lu-DOTATATE PRRT where four of them had proven *SDHB* mutations. Three patients with previously documented progressive disease achieved stabilization; one had partial response and one was treatment refractory. Median progression-free survival was 17 months, which is slightly better than our results in *SDHB* subgroup (12 months). In 2015, Makis et al. [51] described the use of ^177^Lu- DOTATATE in three patients with metastatic disease. They demonstrated partial response and symptomatic and biochemical relief in all three patients. In this study authors reported development a catecholamine crisis and tumor lysis syndrome within hours of treatment, requiring intensive care unit (ICU) support in single patient and similar catecholamine crises in next patient reported few days after therapy. Authors’ experiences with ^177^Lu-DOTATATE treatment suggest that these life–threatening complications in paraganglioma patients may be related to the time of radionuclide administration which should be administered over 2 h at least, but preferably over 4 h, and not over 15–30 min as described in the literature [51].

Indeed, safety of PRRT is an important issue. Most patients with metastatic PPGL are found with a large tumor burden. Therefore, patients with metastatic disease have an elevated risk for severe cardiovascular disease upon exposure to systemic therapies associated with massive release of catecholamines once the tumor destruction starts. In contrast to the paroxysmal symptoms caused by the highly potent hormone adrenaline, these tumors that produce and secrete mainly noradrenaline (Cluster 1 tumors), are commonly associated with a decreased frequency of signs and symptoms related to catecholamine excess [52].

However, it should be emphasized that four of our patients with hormonally active tumors had adequate hypotensive treatment with alpha blockers, amlodipine and/or beta blockers prior to PRRT and none of them had paroxysmal symptoms both before and during PRRT even when level of normetanephrine in plasma was very high (Table 2). It is also noteworthy that, in all our patients, plasma metanephrine levels at presentation was in a normal range. One of the mainstays to increase the safety of PRRT is that all patients with a hormonally functional PPGL should undergo a pretreatment blockade to prevent cardiovascular complications with alpha-adrenergic receptor blockers as the first choice and clinicians should avoid medications that can trigger hemodynamic instability and cardiovascular events (for example, steroids, dopamine D2 receptor antagonists, sympathomimetics, selective serotonin reuptake inhibitors, opioid analgesics, tricyclic antidepressants, and others).

It should be also noted that all patients with hormonally-active tumors had a hormonal response to PRRT. Three of them had a partial response, and in a single case with *SDHD C11X* mutation and inoperable multifocal head and neck PPGLs and middle mediastinum tumor we observed hormone normalization persisting over 10 years of follow-up.

In a published report by Nastos et al. 2 patients with progressive/metastatic PGLs or PCCs were treated with either ^131^I-mIBG, ^90^Y-DOTATATE or ^177^Lu-DOTATATE. Therapy with ^90^Y-DOTATATE resulted in a 100% response rate and patients treated with PRRT had increased PFS and response to treatment compared to ^131^I-mIBG treated patients. These results strongly support use of PRRT in this group of patients [53].

Interesting report was published recently by Kong et al. in 2017 [54]. They assessed the results of PRRT in a series of 22 patients with functional PPGLs. Fourteen patients were treated for uncontrolled HTN and six for progressive or symptomatic metastatic disease or local recurrence. Of the entire cohort, 36% had disease regression on computed tomography, with stable findings in 50%. Median progression free survival was 39 months. Three months after PRRT, eight of 14 patients treated for uncontrolled hypertension required reduced medication doses [54].

In our series of 13 *SDHx* gene mutations carriers median PFS for entire group was 35 months. There was significantly shorter PFS for *SDHB* 12 months compared to *SDHD* not reached, (respectively) which is not astonished since it was previously reported that metastatic SDHB tumors are associated with more aggressive course and worse prognosis. Median overall survival was 68 months, and again similar to PFS, OS was shorter for *SDHB* (25 months) when compared to when *SDHD* was not reached. An interesting finding in our study is that the efficacy of treatment in metastatic PPGLs seems to depend mainly of the presence of liver metastases regardless of the type of mutation. In our datasets with long-term follow-up, those patients with liver lesions had worse prognosis compare to those with isolated bone metastases. While OS for both groups was the same 25 months, the median PFS was shorter in group of liver metastases 10 vs. 12 months (not significant). However, there was a clear benefit in terms of response to treatment in both groups. The favorable effect of PRRT was also noted when we analyzed the performance status (PS both Karnofsky and ECOG/WHO), which was evaluated prospectively six weeks after therapy and then every three months after PRRT.

In the literature we can see that outcome of PRRT in in our homogenous group of *SDHx* mutations carriers is similar to that reported by others in different cohorts of patients. On the other hand, in almost all these reports we can find patients with *SDHx* mutations. In the group of 20 patients of Kong et al., eight had *SDHB* and *SDHD* mutations, two were sporadic, and the genetic background for 10 was unknown [54]. In Nastos et al., a cohort of 22 patients, *SDHB* mutations were found in five cases, genetic status for remaining patients was unknown [53]. Only Zovato and Pinato’s series of four and five patients consisted of “pure” *SDHD* and *SHDB* mutations carriers, respectively [39,41].

It is possible that in all reported series there were more uncovered *SDHx* mutations carries and this fact may affected the outcomes of PRRT. Therefore, it is also possible that effectiveness of PRRT in cluster 1 subgroup *VHL/EPAS1*-related tumors, as well as cluster 2 and cluster 3 tumors may be different. This hypothesis, however, requires further research.

Regarding the adverse events (toxicity) of treatment no documented myelodysplastic syndrome or myelodysplasia was noted in this study. We found severe hematological complication which can be attribute to PRRT only in two cases—in a single subject with *SDHB*, who had a previous history of intensive chemotherapy and radiotherapy, and in another one with *SDHD* and bone marrow involvement. We observed significant G3 anemia after 12 months and after 18 months in a second patient, with normal levels of leucocytes and platelets, which can be associated rather with marrow involvement and progressive disease than with PRRT. Renal failure which required final dialysis occurred in one patient with SDHB in whom we also observed G3 anemia. This particular patient, in addition to earlier chemotherapy and radiotherapy, had four cycles of full-dose ^90^Y DOTATATE PRRT. In the remaining 12 patients we did not observe significant hematological or renal complication.

The limited number of cases limit the strength of our observation. Nevertheless it is still considered a relatively large series and, so far, the largest homogenous series of SDHD and SDHB mutations carriers treated with ^90^Y DOTATATE PRRT.

## 5. Conclusions

This was an important study evaluating the effect of ^90^Y DOTATATE in patients with *SDHD* and *SDHB* mutations with advanced disease.

PRRT represents an active and well tolerated treatment in this group of patients and could be proposed as an available alternative for the treatment of PGL when no other therapy options are available. The importance of such observations comes directly from the concept of personalized medicine, which is especially of value in patients with rare genetic syndromes.

## Figures and Tables

**Figure 1 jcm-08-00952-f001:**
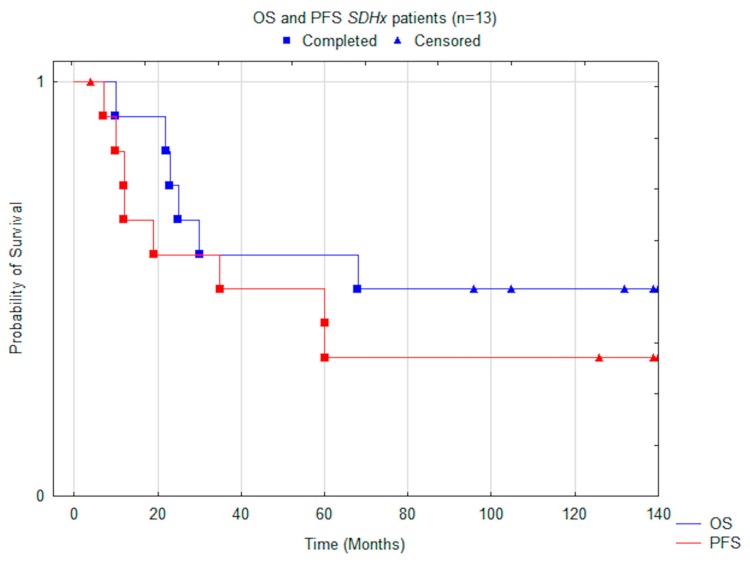
Comparison of median OS (overall survival) and PFS (progression free survival) using the Kaplan–Meier method, including all patients.

**Figure 2 jcm-08-00952-f002:**
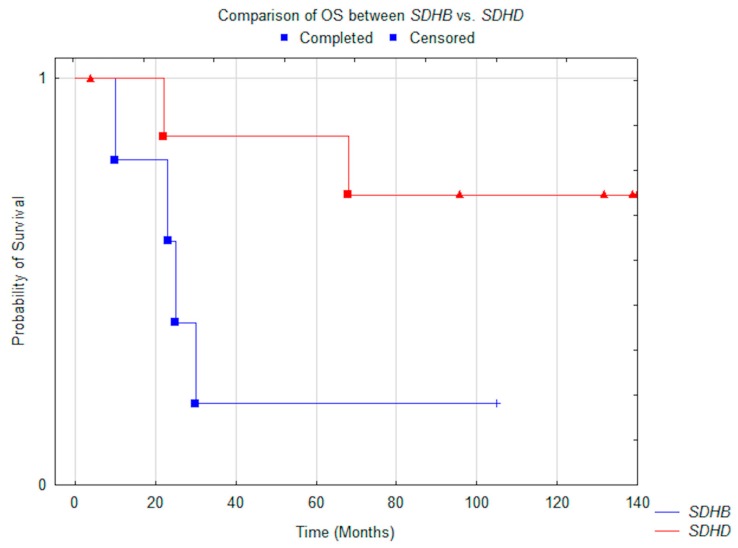
Comparison of median OS between *SDHD* (*n* = 8) vs. *SDHB* (*n* = 5) patients (*p* = 0.05).

**Figure 3 jcm-08-00952-f003:**
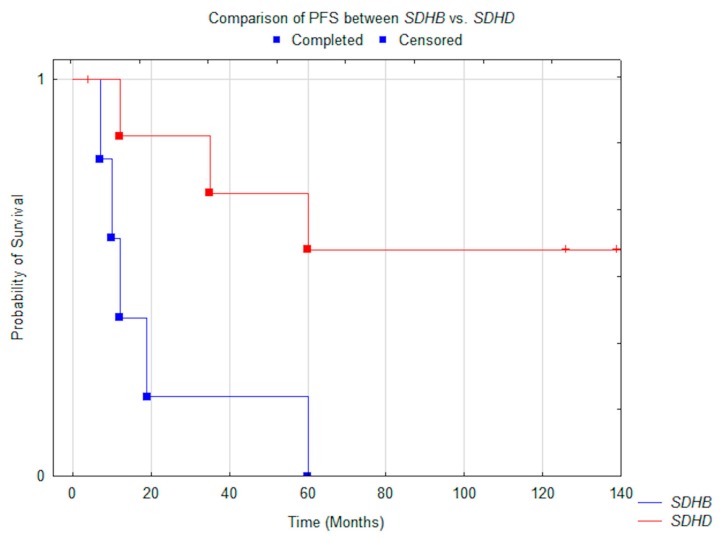
Comparison of median PFS between *SDHD* (*n* = 8) vs. *SDHB* (*n* = 5) patients (*p* = 0.014).

**Figure 4 jcm-08-00952-f004:**
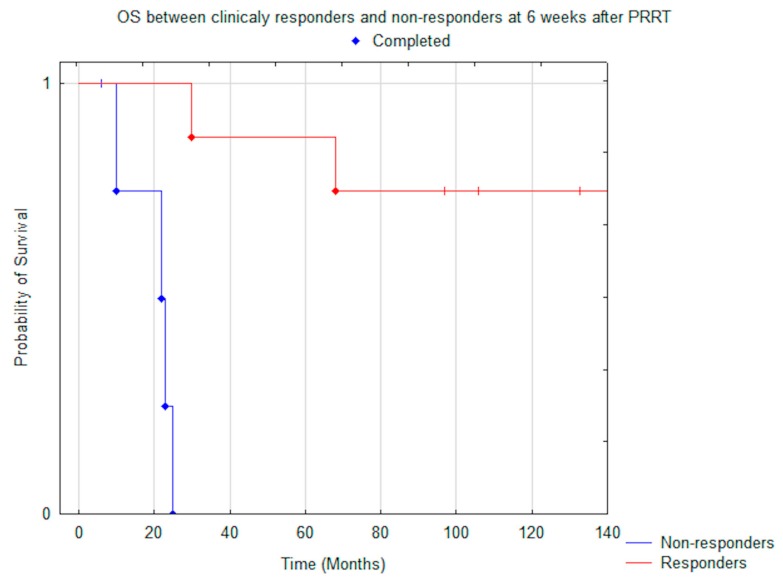
Comparison of median OS between clinically responders (*n* = 8) and non-responders (*n* = 5) at six weeks after PRRT (*p* = 0.005).

**Figure 5 jcm-08-00952-f005:**
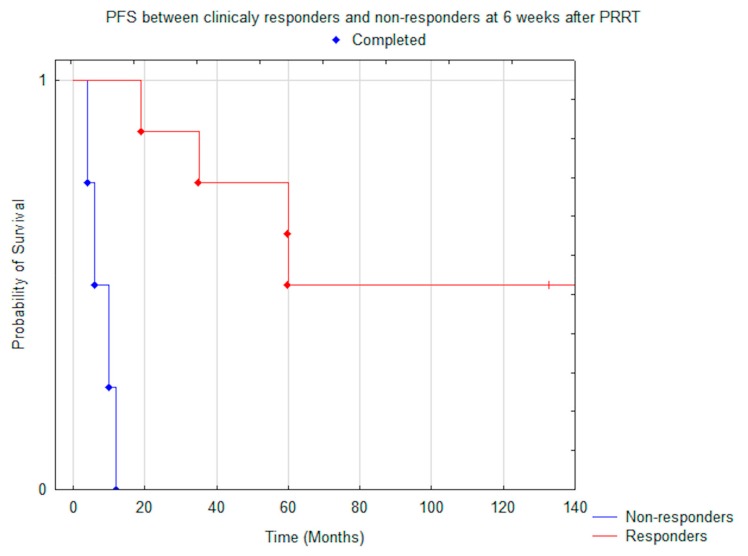
Comparison of median PFS between clinically responders (*n* = 8) and non-responders (*n* = 5) at six weeks after PRRT (*p* = 0.0004).

**Figure 6 jcm-08-00952-f006:**
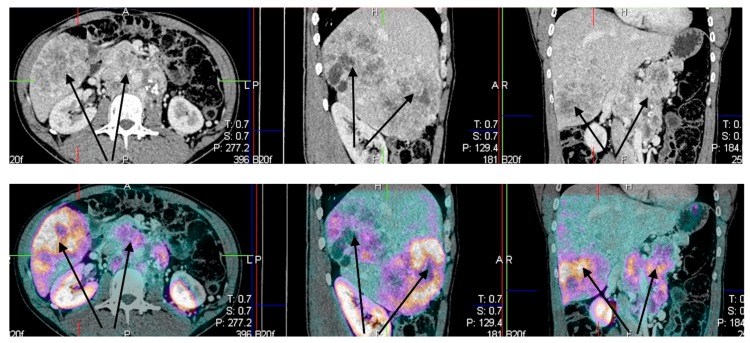
Male, 35 year-old patient with PGL4, standard CT upper panel (**A**) arrows indicated liver and lymph node involvement, and image fusion–lower panel (**B**) SPECT/CT somatostatin receptor scintigraphy (SRS) using 99 mTc HYNICTOC (Tektrotyd^®^, Polatom, PL).

**Table 1 jcm-08-00952-t001:** Baseline demographic, clinical and pathological characteristics for the overall group and the subgroups of patients with SDHD and SDHB mutation.

Variable	All *SDHx*	*SDHD n* = 8	*SDHB n* = 5	*p*-value	
No. of patients, *n* (%)	13 (100)	8 (62)	5 (38)		
Mean age, years (range)	41.9 (27–62)	45.1 (31–62)	36.6 (27–43)	0.13	
Gender, *n* (%)					
Female	5 (38)	4 (50)	1 (20)	0.42	
Male	8 (62)	4 (50)	4 (80)		
Median time from initial diagnosis Months (range)	48.0 (10–180)	114.0 (10–180)	30.0 (18–113)	0.42	
Initial performance status (PS), *n* (%)					
WHO/ECOG = 1	11 (85)	7 (88)	4 (80)	0.09	
WHO/ECOG = 2	2 (15)	1 (12)	1 (20)		
Grading of primary tumor, *n* (%)					
G1	7 (54)	6 (75)	1 (25)		
G2	6 (46)	2 (25)	4 (75)		
Median Ki-67 index, % (95% CI)	2.2 (1.9–6.4)	2.0 (1.6–3.4)	5.0 (0.8–13)	0.048	
Liver involvement *n* (%)	6 (46)	3 (38)	3 (60)	0.83	
Presence of bone mts *n* (%)	9 (41)	4 (50)	5 (100)	0.86	
Secretor tumors, *n* (%)	4 (31)	2 (25)	2 (40)	0.18	
Mean initial CgA x ULN (Range)	5.1 (0.8–26.4)	4.7 (0.78–26.0)	5.8 (1.4–12.0)	0.12	
SRS Krenning uptake scale 3/4	2/11	1/7	1/4	0.97	

The WHO/ECOG performance status (PS) grades the status of patients with respect to activities of daily living on a scale of 0 to 4, with 0 indicating that the patient is fully active. Abbreviations: mts–metastases; CgA–chromogranin A; ULN – upper limit normal.

**Table 2 jcm-08-00952-t002:** Details of genetic mutation in all patients, primary tumor localization, grading (G) of tumor differentiation and mitotic index, as Ki-67, using MIB1 antibody. Additional secretion before PRRT and three months after PRRT.

No	Gender	Age *,	Type of Mutation	Primary Tumor Localization	Grading (Ki-67 in %)	Secretion before PRRT pg/mL	Secretion after PRRT pg/mL (Three Months)
**1**	M	62	*SDHD C11X,* *ex 1 33TGC-TGC*	Right PPC	2 (5)	normal	normal
**2**	F	31	*SDHD C11X,* *ex 1 33TGC-TGC*	Chest & abdominal PGLs	1 (2)	normal	normal
**3**	M	52	*SDHD C11X,* *ex 1 33TGC-TGC*	Right HNP	1 (1)	normal	normal
**4**	F	51	*SDHD C11X,* *ex 1 33TGC-TGC*	Left HNP	2 (3)	MTY = 1890 NMN = 126	MTY = 1060 NMN = 143
**5**	M	42	*SDHD C11X,* *ex 1 33TGC-TGC*	Right HNP	1 (2)	NMN = 183,6	NMN = normal
**6**	F	47	*SDHD C11X,* *ex 1 33TGC-TGC*	Bilateral PPC	1 (2)	normal	normal
**7**	M	31	*SDHD C11X,* *ex 1 33TGC-TGC*	Left PPC	1 (1)	normal	normal
**8**	F	45	*SDHD C11X,* *ex 1 33TGC-TGC*	Left HNP	1 (2)	normal	normal
**9**	M	36	*SDHB ex.3 p.R90X*	Left PPC	1 (2)	MTY = 9189 NMN = 6911	MTY = 7235 NMN = 6430
**10**	M	38	*SDHB* exon 1 deletion	Bladder PGL	2 (5)	normal	normal
**11**	M	27	*SDHB c. 708 T > C* *(int. 574 T > C) heterozygotic*	Paraspinal PGL	2 (8)	MTY = 3344 NMN = 2622	MTY = 2570 NMN = 1987
**12**	F	39	*SDHB exon 1 deletion*	Abdominal PGL	2 (15)	normal	normal
**13**	M	43	*SDHB R230L, exon7*	Left HNP	2 (5)	normal	normal

* Age of patients before PRRT *SDHD*-PGL1 syndrome, *SDHB*–PGL4 syndrome; PPC–pheochromocytoma, PGL–paraganglioma; HNP head and neck paraganglioma; MTY–Methoxytyramine; NMN–Normetanephrine in serum. Plasma NMN (pg/mL) upper reference intervals (age and gender adjusted): patient 4: 147; patient 5: 159.9; patient 9: 115; patient 11: 106; Plasma MN (pg/mL): 88, Plasma MTY (pg/mL): 30.

**Table 3 jcm-08-00952-t003:** History of previous treatment for the overall group and subgroups of patients with SDHD and SDHB mutations.

Variable	All patients (*n* = 13)	*SDHD* (*n* = 8)	*SDHB* (*n* = 5)	*p*-Value
Previous surgery, *n* (%)	13 (100)	8 (100)	5(100)	
Initial Previous surgery with ITT (intention to treat) *n* (%)	5 (38)	3 (38)	2 (40)	0.68
Previous SST analogues, *n* (%)	8 (62)	6 (75)	2 (40)	0.56
Previous any other systemic or local therapy (%) #	9 (69)	5 (63)	4 (80)	0.42

# chemotherapy, external beam radiotherapy, embolization, 131ImIBG therapy, thermoablation.

**Table 4 jcm-08-00952-t004:** Details of given activity in group all SDHx patients and additional in SDHD and SDHB subjects.

	All (*n* = 13)	PGL-1 (*n* = 8)	PGL-4 (*n* = 5)
Mean therapy sessions	2.5	2.4	3.0
Activity per session ^90^Y in GBq mean (range)	3.4	2.9	3.3
Cumulative Activity ^90^Y in GBq, mean (range)	8.3	7.3	9.9

**Table 5 jcm-08-00952-t005:** Median overall survival (OS) and progression-free survival (PFS) for the overall group of patients with SDHx mutation including both subgroups SDHD and SDHB, hormonal status in entry into the study, presence of liver and bone involvement and also clinical response evaluated six weeks after PRRT as partial response (PR), stable disease (SD) or disease progression (PD). Abbreviation: N.R. not reached.

Variable	Subjects	Median OS Months (+/−95% CI)	*P*-value	Median PFS Months (+/−95% CI)	*p*-Value
All subjects	13	68.0 (38.6–105.1)		35.0 (24.4–93.1)	
Mutation			0.05		0.014
SDHD	8	N.R. (not reached)		N.R. (not reached)	
SDHB	5	25.0 (3.2–85.6)		12.0 (2.3–48.8)	
Clinical response			0.005		0.0004
PR	8	N.R. (not reached)		N.R. (not reached)	
SD/DP	5	22.0 (6.5–27.9)		7.0 (3.8–11.8)	
Liver mts			0.033		0.005
present	7	25.0 (6.0–67.7)		10.0 (1.3–39.0)	
absent	6	N.R. (not reached)		N.R. (not reached)	
Bone mts			0.029		0.0027
present	9	25.0 (14.1–71.9)		12 (6.3–41.0)	
absent	4	N.R. (not reached)		N.R. (not reached)	
Hormonal status			0.496		0.84
secretor	4	49.0 (0–152.4)		27.0 (0–148.2)	
non-secretor	9	97.0 (30.0–120.1)		60.0 (15.9–109.4)	

**Table 6 jcm-08-00952-t006:** Distribution of ORR based on RECIST 1.0-radiological responses for the overall group and for the subgroups of patients with PGL1 (SDHD) or PGL4 (SDHB) NENs.

Variable	All *n* (%)	*SDHD**n* (%)	*SDHB n* (%)
RECIST six months,	*n* = 12	*n* = 8	*n* = 4
PR	1 (8)	0	1 (25)
SD	9 (75)	6 (75)	3 (75)
DP	2 (17)	2 (25)	0
RECIST 12 months,	*n* = 11	*n* = 8	*n* = 4
PR	0	0	0
SD	9 (82)	6 (75)	3 (75)
DP	2 (18)	2 (25)	1 (25)

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
