# Peer review of "A Clinical Efficacy of PRRT in Patients with Advanced, Nonresectable, Paraganglioma-Pheochromocytoma, Related to SDHx Gene Mutation"

_jcm, 2019, doi:10.3390/jcm8070952_

Round 1
Reviewer 1 Report
The authors conducted a study evaluating the efficacy of PRRT in patients with SDHx mutations that have advanced pheochromocytoma/paragangliomas. This is an important study that adds much needed information to the literature about PRRT therapy in this disease process.
However, there are several areas that need further clarification:
1.In the abstract, the authors may consider mentioning that this is a prospective, open label phase II study.
2. Introduction line 68, the TCGA project actually identified 4 Clusters. The
fourth cluster is the cortical admixture type that has the MAX mutation
associated with it.
3. Introduction lines 102-103, “Furthermore, cluster 1 pseudohypoxic TCA
cycle-related PPGLs show a relatively well cell differentiation, often with a decreased or absence of key enzyme in catecholamine metabolism what results in purely noradrenergic…” do the authors mean the cells are de-differentiated?
4. Introduction lines 137-138, “Our series consists of 13 subjects with PGL1
and PGL4, all with metastatic or unresectable PPGLs is so far the largest
one.” There are several studies such as references 40 and 54 that have a
larger series of patients that underwent PRRT.
5. How many patients evaluated for this study had SDHx mutations but
negative SST expression?
6. After PRRT treatment, there was a some trend in lower catecholamines but
not a profound drop. Did the authors need to reduce the anti-hypertensive
medications for these patients?
7. There were 12 patients that had radiographical assessment. What happened
to the 13th patient?
8. The authors may want to comment on why in their trial they experienced no
hypertensive crisis compared to other studies. Perhaps a technique the
authors are using may help other groups who are performing PRRT.
9. In the Discussion, lines 484-485, “While OS for both groups was the same 25
months, the median PFS was shorter in group of liver mts 10 vs. 12 months
(P>0.05).” This seems like a small difference, how did this make statistical
significance?
10. Although, in some parts of the world certain words are spelled in a specific
way and expressed a little differently, the authors may consider revising
grammar and spelling. Here are a few examples:
-mIBG should be MIBG;
-in Table 2- (3M)- may need to label as “M”-month;
-“N.R.”- there is no definition that was easily available to see
-metanephryne- metanephrine
-Under “Results” lines 296-298; “During and after PRRT anyone subjects had significant clinical symptoms and signs due to catecholamine oversecretion, which were presented before treatment, which consists of increased blood pressure, increase heart rate etc “- unclear what this means.
-Under “Discussion,” lines 491-492: “On the other hand, in almost all these reports we can find patients with SDHx mutations.” It is unclear as to what the precise message is.
-May consider spelling out PRRT in the abstract.
-In the Introduction- on the first line, paragangliomas arise from the sympathetic/parasympathetic chain ganglia. Pheochromocytomas arise from chromaffin cells.
-Ref 53, “treatment” is misspelled.
-The authors may want to add arrows to Figure 6A and 6B.
Author Response
Comments and Suggestions for Authors
The authors conducted a study evaluating the efficacy of PRRT in patients with SDHx mutations that have advanced pheochromocytoma/paragangliomas. This is an important study that adds much needed information to the literature about PRRT therapy in this disease process.
However, there are several areas that need further clarification:
1. In the abstract, the authors may consider mentioning that this is a prospective, open label phase II study. Is corrected and information put in the abstract.
2. Introduction line 68, the TCGA project actually identified 4 Clusters. The fourth cluster is the cortical admixture type that has the MAX mutation associated with it. Thank you for this comment. We added in the Introduction section:
“Finally, cluster 4 - the cortical admixture subtype overexpressed known adrenal cortex markers (CYP11B1, CYP21A2 and STAR) and PPGLs markers and is associated with mutations in MAX.”
3. Introduction lines 110-111, “Furthermore, cluster 1 pseudohypoxic TCA cycle-related PPGLs show a relatively well cell differentiation, often with a decreased or absence of key enzyme in catecholamine metabolism what results in purely noradrenergic…” do the authors mean the cells are de-differentiated?
In pheochromocytoma/paraganglioma tumors variable expression of biosynthetic enzymes, highly influenced by the underlying mutation, leads to profound differences in the types and amounts of catecholamines being produced. This is accomplished by mutation-dependent differentiation of progenitor cells, which regulates the expression of enzymes involved in the synthesis of catecholamines. It means that tumor progenitor cells which give rise to PPGLs with the adrenergic phenotype may be more differentiated than the noradrenergic phenotype, which in turn, may be more differentiated than the dopaminergic phenotype. There is a mistake in the sentence quoted by the reviewer which we have corrected. Instead “a relatively well cell differentiation” should be “a relatively low cell differentiation”.
4. Introduction lines 137-138, “Our series consists of 13 subjects with PGL1 and PGL4, all with metastatic or unresectable PPGLs is so far the largest one.” There are several studies such as references 40 and 54 that have a larger series of patients that underwent PRRT.
There are only two reports in the literature concerned outcome of PRRT in patients with exclusively SDHx mutations; Zovato et al. described 4 patients with PGL1 and head and neck paragangliomas , Pinato et al. described 5 patients with PGL4 and metastatic PPGLs. In this context our series consists of 13 subjects purely with SDHx mutations, is indeed so far the largest one.
5. How many patients evaluated for this study had SDHx mutations but negative SST expression?
All patients had positive SSTR expression, it was worth to include to the study using PRRT as therapy option. All of them had increase SST receptor expression Krenning scale at least 2 but our all patients had uptake grade 3 or 4.
6. After PRRT treatment, there was a some trend in lower catecholamines but not a profound drop. Did the authors need to reduce the anti-hypertensivemedications for these patients?
In contrast to the paroxysmal symptoms caused by the highly potent hormone epinephrine, cluster 1 pseudohypoxic TCA cycle-related tumors that produce and secrete mainly norepinephrine, are commonly associated with a decreased frequency of signs and symptoms related to catecholamine excess and present more commonly with non-episodic hypertension. In only one patient with normalization of plasma free catecholamines after PRRT we were able to withdraw all antihypertensive drugs. The remaining patients required further antihypertensive treatment.
7. There were 12 patients that had radiographical assessment. What happened to the 13th patient? This particular SDHB mutation patient (PGL-4), had primary bladder PGL and multiple bone mts. The patient had only a single PRRT treatment session and then he developed a pathological spine fracture, which led to tetraplegia. After the fracture, the patient was unable to reach the next dose of PRRT. He was in the of BSC treatment for the next year. He died 12 months after giving PRRT.
We added a paragraph to the Radiological Response section: One SDHB mutation carrier with urinary bladder tumor and bone metastases had only a single PRRT treatment session and then he developed a pathological spine fracture, which led to tetraplegia. After the fracture, the patient was unable to reach the next dose of PRRT and died 12 months later .
8. The authors may want to comment on why in their trial they experienced no hypertensive crisis compared to other studies. Perhaps a technique the authors are using may help other groups who are performing PRRT.
We believe that three basic factors play a fundamental role in the safety of PRRT:
1. We think that in noradrenergic tumors (like in cluster 1 pseudohypoxic TCA cycle-related tumors) the PRRT treatment may be safer that in adrenergic tumors. As we mentioned above, epinephrine is a highly potent circulating hormone and patients with PPGLs secreting mainly epinephrine more frequently that in noradrenergic PPGLs have paroxysmal symptoms of hypertension, palpitations, anxiety, flushing and sweating which can be additionally triggered by PRRT treatment.
We added to the discussion: “It means that PRRT treatment may be safer in pure noradrenergic tumors (like in our group with SDHx mutations) than in those tumors which excrete adrenaline. Adrenaline is a highly potent circulating hormone and patients with PPGLs secreting mainly epinephrine more frequently that in noradrenergic PPGLs have paroxysmal symptoms of hypertension, palpitations, anxiety, flushing and sweating which can be additionally triggered by PRRT treatment.”
2. One of the mainstays to increase the safety of PRRT is that all patients with a hormonally functional PPGL should have adequate antihypertensive treatment with alfa-adrenergic receptor blockers as the first choice and medications that can trigger haemodynamic instability should be avoided and we discuss this issue in the Discussion section.
3. One of the most important for the safety of PRRT is the technique. In our administration protocol, 8 mg of Ondansetron was administered i.v. 30 min before amino acid (AA) administration. The AA infusion was to prevent the development of delayed renal toxicity. The slow intravenous infusion of AA were carried out for 1.0 to 1.5 hours, then 90Y DOTATATE was added via an infusion pump system with mean speed 150ml/h and was continued for a approx. of 20-25 minutes. Every subject was treated at intervals 8-12 weeks.
9. In the Discussion, lines 494-495, “While OS for both groups was the same 25 months, the median PFS was shorter in group of liver mts 10 vs. 12 months (P>0.05).” This seems like a small difference, how did this make statistical significance? Corrected as n.s. not significant.
10. Although, in some parts of the world certain words are spelled in a specific way and expressed a little differently, the authors may consider revising grammar and spelling. Here are a few examples:
· mIBG should be MIBG; we do not agree with this. The right term of mIBG is meta-IodoBenzylGuanidyne (mIBG), prefix m as meta is the same as metastable technetium 99mTc.
in Table 2- (3M)- may need to label as “M”-month corrected;
“N.R.”- there is no definition that was easily available to see – will be corrected and explained as not reached.
metanephryne- metanephrine - corrected
Under “Results” lines 305-307; “During and after PRRT anyone subjects had significant clinical symptoms and signs due to catecholamine oversecretion, which were presented before treatment, which consists of increased blood pressure, increase heart rate etc “- unclear what this means.
Thank you for this very helpful comment. We corrected this as follows: “During and after PRRT anyone subjects had significant clinical symptoms and signs due to catecholamine oversecretion.”
Under “Discussion,” lines 491-492: “On the other hand, in almost all these reports we can find patients with SDHx mutations.” It is unclear as to what the precise message is.
We are grateful for this comment. We would like our message to be clear to the reader so we added to the discussion section: “ In the literature we can see that outcome of PRRT in our homogenous group of SDHx mutations carriers is similar to that reported by others in different cohorts of patients. On the other hand, in almost all these reports we can find patients with SDHx mutations which may affect the results of treatment.”
May consider spelling out PRRT in the abstract – added and corrected.
In the Introduction- on the first line, paragangliomas arise from the sympathetic/parasympathetic chain ganglia. Pheochromocytomas arise from chromaffin cells.
Thank you for this very helpful suggestion. We added in the Introduction section: “ Paragangliomas arise from the sympathetic/parasympathetic chain ganglia. Pheochromocytomas arise from adrenal medulla chromaffin cells.”
Ref 53, “treatment” is misspelled. – corrected.
The authors may want to add arrows to Figure 6A and 6B – done.
Reviewer 2 ReportAn interesting prospective evaluation of unresectable PGLs by PRRT is presented form a polish single institution. The project is funded by a polish research fund. The treatment of advanced PGL with metastatic and aggressive clinical course is a real challenge. The approach is always multidisciplinary.
Major Requests:
The Introduction section should be shortened and more focused on the therapeutic tool PRRT. The rationale of it should be explained to the reader (line 129-134). Maybe the genetic part could be shortened.
Material and Methods section needs spell and grammar check as well as professional English editing. Furthermore the exact time span of the study is not given in the text.
Major parts of the material and methods section should be transferred in the results section. A clear definition of inclusion and exclusion criteria is missing in material and methods.
Table 5: How do you create p-values? Many variables are given as N.R., which is not explained in the figure legend.
Minor Request
The Affiliation of the some authors are not clear? Departements?
The Abstract should be structured
line 110: it has been shown
line 137/138: grammar check needed
Table 1: Abbreviations should be explained in detail in the figure legend
Table 3: Please explain ITT and SST in this context!
The Discussion section is comprehensive, but also needs English editing.
Unfortunately the whole manuscript is not very well structure and the English spelling is not suitable for publication in an international journal. Thus the manuscript should be rearranged and edited by a native speaker.
As this was funded research it may be particularly interesting if a multicentric approach is planned for further evaluation of PRRT?
The authors should explain if the study is registered in a local or international study registry, e.g. trials.gov or similar?
Author Response
Comments and Suggestions for Authors
An interesting prospective evaluation of unresectable PGLs by PRRT is presented form a polish single institution. The project is funded by a polish research fund. The treatment of advanced PGL with metastatic and aggressive clinical course is a real challenge. The approach is always multidisciplinary.
Major Requests:
The Introduction section should be shortened and more focused on the therapeutic tool PRRT. The rationale of it should be explained to the reader (line 129-134). Maybe the genetic part could be shortened.
The main goal of our research was to evaluate the effectiveness of PRRT treatment in the group of patients with SDHx gene mutations. Among genetic alterations in PPGLs, mutations in succinate dehydrogenase (SDHx) predispose patients to aggressive phenotypes, such as distal metastasis, in some series in up to 70% of patients, and tumor multiplicity and recurrence, which emphasizes an urgent need for effective therapies against SDHx-mutated PPGLs. Taking into consideration that among hereditary PPGLs, SDHx syndromes are most common, our study was focused on this special group. Data from transcriptomic studies indicates that SDHx syndromes have a unique molecular-clinical-biochemical-imaging phenotype which can be used in personalized management involving diagnostics, targeted therapy and follow-up. Noradrenergic phenotype, significantly high expression of SSTRs can determine both the potential effectiveness of PRRT therapy and its safety and we describe these phenomena in the introduction and then discuss it in a Discussion section. PRRT is a well-known therapeutic tool used in any neuroendocrine tumors and the technique used in our study was standard and previously described in the literature. These are the reasons why the main emphasis in the introduction was put on genetic and transcriptomic aspects. We shortened the introduction section.
Material and Methods section needs spell and grammar check as well as professional English editing. Furthermore the exact time span of the study is not given in the text.
Thank you for this helpful comment. We added time span of the study in the Materials and Methods section.
Major parts of the material and methods section should be transferred in the results section. A clear definition of inclusion and exclusion criteria is missing in material and methods.
We restructured the manuscript and transferred parts of the material and methods section in the results section. We also added inclusion and exclusion criteria.
Table 5: How do you create p-values? Many variables are given as N.R., which is not explained in the figure legend.
Thank you for pointed out this vague abbreviation – N.R means “not reached” and we added it to the text. The P value was created using the Kaplan-Meier method, the differences between groups in both PFS and OS were calculated using Cox-Mantel test. All statistical calculations were carried out using Statistica 12 package (StatSoft, Tulsa OK, USA), which is mentioned in Statistical analysis.
Minor Request
The Affiliation of the some authors are not clear? Departements?
We corrected this.
The Abstract should be structured it is done as editor request we corrected as requested first reviewer and try structured as editor requested
line 110: it has been shown – see below
line 137/138: grammar check needed see below
We decided to send a manuscript to proofread the English language.
Table 1: Abbreviations should be explained in detail in the figure legend
We added abbreviations in the figure 1 legend.
Table 3: Please explain ITT and SST in this context!
Thank you for your comment. Both abbreviations are explained in the text below table. ITT – intention to treat and SST means – somatostatin.
The Discussion section is comprehensive, but also needs English editing.
We decided to send a manuscript to proofread the English language.
Unfortunately the whole manuscript is not very well structure and the English spelling is not suitable for publication in an international journal. Thus the manuscript should be rearranged and edited by a native speaker.
As this was funded research it may be particularly interesting if a multicentric approach is planned for further evaluation of PRRT?
We currently plan a prospective multicenter national trial using 177Lu DOTATATE or 177Lu DOTATOC in similar group of patients.
The authors should explain if the study is registered in a local or international study registry, e.g. trials.gov or similar?
We are currently under final evaluation of registration in www.clinicaltrials.gov. We will notify you immediately after acceptance and giving the number in Clinicaltrials.
Round 2
Reviewer 1 Report
This is a revision of the manuscript where the authors describe the efficacy of PRRT therapy in patients with pheochromocytoma/paraganglioma with SDHx mutations.
There remains a few points of clarification:
1.It was previously asked if the authors saw a reduction in anti-hypertensive meds required for those patients who were found to have lower catecholamines. The answer was that they required further hypertensive therapy. To further clarify the question, did the authors need to reduce dosages or the number of anti-hypertensive medications?
2.In the “Introduction,” lines 78-79, “Transcriptomic profiling has identified 3 clusters...” This needs to be corrected to 4 clusters.
3.In the “Introduction,” lines, 84-87, “ Cluster 2 includes Wnt signaling…Cluster 3 tumors develop in patients with … RET..” It is actually reversed, Cluster 2 is associated with kinase signaling pathway.
4.“Clinical Response,” line 321, “…2 subjects with PGL1 and 3 with PGL4 had serum hormone metabolite reduction…”
This statement does not match the data on Table 2.
5.In “Conclusion,” “This is the first study evaluating the effectiveness of
PRRT…” The authors may want to rephrase by saying that this was an important study evaluating the effect of 90Y DOTATATE in patients with SDHD and SDHB mutations with advanced disease.
6. Although this version is much improved in terms of spelling and grammar,
more effort needs to be placed in this. These are just a few exmaples:
a.Under “Biodistribution of the radiotracer” line 212 “Bremsstrahlung”
b.Line 322, under “Clinical Response,” – metanephrine spelling
c.“Discussion,” line 422, “PPRT in 13 patients,” should be PRRT.
d.“Discussion,” spelling- alpha blockers
Author Response
Response to Reviewer 2
This is a revision of the manuscript where the authors describe the efficacy of PRRT therapy in patients with pheochromocytoma/paraganglioma with SDHx mutations.
There remains a few points of clarification:
1. It was previously asked if the authors saw a reduction in anti-hypertensive meds required for those patients who were found to have lower catecholamines. The answer was that they required further hypertensive therapy. To further clarify the question, did the authors need to reduce dosages or the number of anti-hypertensive medications?
In only one patient with moderate plasma normetanephrine elevation before and normalization after PRTT we were able to withdraw all antihypertensive drugs. The remaining 3 patients initially had a significantly elevated level of plasma metanephrines with only slight decreased after PRRT and the antihypertensive treatment after PRRT consisted of the same medications in the same doses.
We added to the legend of the table 2 ranges of normal values for plasma metanephrines and following paragraph to Clinical Response section. “In all patients with secretor tumours, (2 subjects with SDHD and 2 with SDHB) we observed early hormonal response. In only one patient with moderate plasma normetanephrine elevation before and normalization after PRTT we were able to withdraw all antihypertensive drugs. The remaining 3 patients initially had a significantly elevated level of plasma metanephrines with only slight decreased after PRRT and the further antihypertensive treatment consisted of the same medications in the same doses”.
2. In the “Introduction,” lines 78-79, “Transcriptomic profiling has identified 3 clusters...” This needs to be corrected to 4 clusters. It was corrected.
3. In the “Introduction,” lines, 84-87, “Cluster 2 includes Wnt signaling…Cluster 3 tumors develop in patients with … RET..” It is actually reversed, Cluster 2 is associated with kinase signaling pathway. It was corrected.
4. “Clinical Response,” line 321, “…2 subjects with PGL1 and 3 with PGL4 had serum hormone metabolite reduction…”This statement does not match the data on Table 2.
Thank you very much for indicating the error. We apologize for this mistake, we corrected this.
5. In “Conclusion,” “This is the first study evaluating the effectiveness of PRRT…” The authors may want to rephrase by saying that this was an important study evaluating the effect of 90Y DOTATATE in patients with SDHD and SDHB mutations with advanced disease.
Thank you for this very helpful comment. It is much better. We have introduced this amendment at the end of the discussion.
6. Although this version is much improved in terms of spelling and grammar, more effort needs to be placed in this. These are just a few exmaples:
a. Under “Biodistribution of the radiotracer” line 212 “Bremsstrahlung”
It was corrected
b. Line 322, under “Clinical Response,” – metanephrine spelling
It was corrected.
c. “Discussion,” line 422, “PPRT in 13 patients,” should be PRRT.
It was corrected.
d. “Discussion,” spelling- alpha blockers
It was corrected.
We hope that the re-revised manuscript is now acceptable.
Yours sincerely,
Jarosław B. Ćwikła